# Error Analysis and Modeling for an Absolute Capacitive Displacement Measuring System with High Accuracy and Long Range

**DOI:** 10.3390/s19245339

**Published:** 2019-12-04

**Authors:** Dongdong Zhang, Li Lin, Quanshui Zheng

**Affiliations:** 1State Key Laboratory of Tribology, Tsinghua University, Beijing 100084, China; zdd15@mails.tsinghua.edu.cn (D.Z.); zhengqs@tsinghua.edu.cn (Q.Z.); 2Department of Mechanical Engineering, Tsinghua University, Beijing 100084, China; 3Center for Nano and Micro Mechanics, Tsinghua University, Beijing 100084, China; 4Department of Engineering Mechanics, Tsinghua University, Beijing 100084, China

**Keywords:** absolute displacement measurement, nanometer accuracy, long range, error analysis

## Abstract

We proposed a novel kind of absolute capacitive grating displacement measuring system with both high accuracy and long range in a previous article. The measuring system includes both a MOVER and a STATOR, the contact surfaces of which are coated by a thin layer of dielectric film with a low friction coefficient and high hardness. The measuring system works in contact mode to minimize the gap changes. This paper presents a theoretical analysis of the influence of some factors, including fabrication errors, installation errors, and environment disturbance, on measurement signals. The measuring signal model was modified according to the analysis. The signal processing methods were investigated to improve the signal sensitivity and signal-to-noise ratio (SNR). The displacement calculation model shows that the design of orthogonal signals can solve the dead-zone problem. Absolute displacement was obtained by a simple method using two coarse signals and highly accurate displacement was further obtained while using two fine signals with the help of absolute information. According to the displacement calculation model and error analysis, the error in fine calculation functions mainly determines the model’s accuracy and is locally affected by coarse calculation functions. It was also determined that amplitude differences, non-orthogonality, and signal offsets are not related to the accuracy of the displacement calculation model. The experiments were carried out to confirm the abovementioned theoretical analysis. The experimental results show that the displacement resolution and error in the displacement calculation model reach ±4.8 nm and ±34 nm, respectively, in the displacement range of 5 mm. The experiments and the theoretical analyses both indicate that our proposed measuring system has great potential for achieving an accuracy of tens of nanometers and a range of hundreds of millimeters.

## 1. Introduction

Displacement measurement with nanoscale resolution and accuracy in the range of several hundred millimeters is crucial in many industrial fields, including semiconductor manufacturing and ultra-precision machining [1,2,3,4]. It is very challenging to achieve both high-accuracy and long-range displacement measurements [5,6]. Among the various kinds of displacement sensors [7,8,9,10,11], laser interferometers, grating rulers, and capacitive grating sensors are the most commonly used types of transducers for displacement measurement, with comparable accuracy over a relatively long range [12,13,14,15,16,17,18]. Laser interferometers have a range of dozens of centimeters, or even several meters, with an accuracy of greater than ±0.1 ppm [13,19,20,21]. In addition to their cost and complicated structure, they are sensitive to many factors, including beam interference, optical mixing, air temperature and humidity, and variation in the optical medium [22,23,24]. They are only suitable for use in well-controlled environments, such as those in calibration applications, due to these drawbacks [5,17,25,26]. When compared with laser interferometers, grating rulers are less susceptible to the environment, and they are universally used in workshop situations that require high accuracy and a long range. The accuracy of commercial grating rulers is usually about ±1 μm, and it is quite difficult to improve the accuracy due to the restrictions of nanofabrication [27,28]. Due to the advantages of a simple structure, low cost, low power consumption, and robustness to the environment [8,29,30], capacitive grating displacement sensors have been arranged with periodical electrodes to achieve both high precision and a long range [31,32,33]. The measurement precision of such area-change-based capacitive grating sensors is very vulnerable to gap changes [16,26,34]. A contact-type sensor was proposed for reducing errors in gap change, but it was not able to truly reach the goal of long-range measurement due to the dead-zone regions that exist in periodic signals where measurement is insensitive to changes in displacement, which results in a major accuracy problem [35,36]. Another kind of time-grating-based sensor was reported in which the resolution is not limited by the electrode pitch and gap. However, signal qualities, including amplitude differences, phase differences, and the offset caused by errors in fabrication and installation, affect the accuracy of these sensors [17,25,37,38].

Absolute displacement sensors immediately provide absolute position information without searching for references through motion under the condition of rebooting after a power loss [39,40]. Absolute displacement sensors remove cumulative errors and provide position information more efficiently when compared with incremental sensors [41]. These characteristics of absolute displacement sensors are essential to closed-loop feedback control in industrial production. Generally, laser interferometers obtain the absolute position while using the time-of-flight method or the multi-wavelength method; however, these methods are very complex [42,43,44]. Absolute grating rulers or grating encoders apply binary code patterns that occupy one or more code tracks for obtaining absolute displacement information [45]; however, these code tracks are difficult to manufacture due to accuracy requirements [46]. It is also difficult and time-consuming to obtain absolute position information from grating rulers or grating encoders [47,48]. There are very few reports on the methods for capacitive grating linear displacement sensors, and the precision of the periodic size of electrodes according to calculation principles greatly influences the accuracy of an absolute displacement measurement [49,50].

Previously, we proposed an absolute capacitive grating displacement measuring system with both high accuracy and long range that includes a MOVER and a STATOR [16]. A thin layer of dielectric film with a low friction coefficient is coated on the contact surfaces of the MOVER and the STATOR. The measuring system works in the contact mode to minimize the gap changes, and the measurement accuracy hardly suffers from the non-uniformity in the gap when the MOVER moves relative to the STATOR. A simple and novel method for obtaining the absolute displacement was introduced into the measuring system, and this method ensures that the measuring system’s accuracy is almost unaffected by errors in the fabrication and installation. Dead-zone regions are first pointed out and two orthogonal periodic signals are then selectively and alternately used to solve the problem.

In this paper, a signal model is constructed based on measuring principles and further modified by taking the influences of fabrication errors, installation errors, and environment disturbances into account. The signal processing methods are investigated to improve the signal sensitivity and resolution. A displacement calculation model is established to obtain the absolute displacement with high accuracy and provide theoretical support to an error analysis. The error analysis of the displacement calculation model identified the major sources of error in the calculation model and provided information that was used to increase the model’s accuracy. The error in the displacement calculation model is unrelated to signal amplitude difference, non-orthogonality, and signal offset, according to the displacement calculation model and the error analysis. The final section describes the experiments and analyses that were carried out.

## 2. Basic Measuring Principle

Figure 1a shows the overall structure of the proposed displacement measuring system, which includes a MOVER and a STATOR [16]. The MOVER and the STATOR both consist of periodically arranged electrodes that were covered with a thin layer of dielectric film with a low friction coefficient (Figure 1a,c). The MOVER moves relative to the STATOR in the contact mode along the X direction. There are four rows of metal electrodes on the MOVER: two rows labeled as M_c_ with only one electrode in each row and another two rows of metal electrodes labeled as M_f_ with *n* (*n* = 3) electrodes in each row. The width and length of the electrodes in M_f_ are $W_{fg}$ and $L_{fg}$, respectively, and the interval between two adjacent electrodes is also $W_{fg}$. The width and length of the electrodes in M_c_ are $W_{cg}$ and $L_{cg}$, respectively. All of these electrodes on the MOVER are connected together. On the STATOR, the four grating-pattern groups of electrodes that are labeled A, B, C, and D in the middle two rows perform fine measurements, and the four grating-pattern groups of electrodes labeled E, F, G, and H in the bilateral two rows perform coarse measurements to provide absolute displacement information. The electrode width is $W_{f}$, the electrode length is $L_{f}$, and the interval between two adjacent electrodes is $I_{f}$, as shown in Figure 1b. The two rows of electrodes are offset by a distance of $W_{fg}/2$, and $W_{fg}$ is equal to ($W_{f}\;$+$\;I_{f}$). The four grating-pattern groups A, B, C, and D in combination with the electrodes in M_f_ form the variable capacitor groups (VCGs), labeled $C_{A}$, $C_{B}$, $C_{C}$, and $C_{D}$, respectively, and these four VCGs are used to perform fine measurements. As shown in Figure 1d, the electrode width is $W_{c}$, the electrode length is $L_{c}$, and the interval between two adjacent electrodes is $I_{c}$. There is also a difference in the distance of $W_{cg}/2$ between the two rows of coarse electrodes, and $W_{cg}$ is equal to ($W_{c}\;$+$\;I_{c}$). The four grating-pattern groups E, F, G, and H in combination with the electrodes in M_c_ also form the variable capacitor groups (VCGs), labeled $C_{E}$, $C_{F}$, $C_{G}$, and $C_{H}$, respectively, and these four VCGs are used to perform coarse measurements.

For an ideal parallel-plate capacitor, the gap between two electrode plates, the overlapping area of the two plates, and the dielectric properties of the insulator between the plates determine the capacitance. As is shown in Figure 1, according to C=εS/d, $C_{A}\left(x\right)$ is described as:(1)$$C_{A}\left(x\right)=\{\begin{array}{ll} \frac{n\varepsilon }{d}\left(W_{f}-x\right)L_{f}+C_{A}^{0}, & x\in \left[0,W_{f}\right)\\ C_{A}^{0}, & x\in \left[W_{f},W_{fg}\right)\\ \frac{n\varepsilon }{d}\left(x-W_{f}-I_{f}\right)L_{f}+C_{A}^{0}, & x\in \left[W_{fg},2W_{fg}-I_{f}\right)\\ \frac{n\varepsilon }{d}W_{f}L_{f}+C_{A}^{0}, & x\in \left[2W_{fg}-I_{f},2W_{fg}\right] \end{array}$$
where ε is the permittivity of the dielectric materials between the metal electrodes on the MOVER and the STATOR, *d* is the distance between the metal electrodes on the MOVER and the STATOR, *n* is the number of electrodes M_f_ in each row on the MOVER, and *x* is the relative displacement of the MOVER and the STATOR in the X direction. The constant $C_{A}^{0}$ is introduced into Equation (1) due to the parasitic capacitance.

To simplify Equation (1), it can be rewritten as:(2)$$C_{A}\left(x\right)=C_{f}\left(x\right)+a$$
where
(3)$$C_{f}\left(x\right)=\{\begin{array}{c} \frac{-2A_{f}}{W_{f}}x+A_{f}\\ -A_{f}\\ \frac{2A_{f}}{W_{f}}x-A_{f}\left(3+\frac{2I_{f}}{W_{f}}\right)\\ A_{f} \end{array}$$
(4)$$a=A_{f}+C_{A}^{0}.$$

In Equations (3) and (4), $A_{f}=\frac{n\varepsilon W_{f}L_{f}}{2d}$ is the amplitude of the functions for the fine measurement. In fact, $C_{f}\left(x\right)$ is a periodic function within the displacement range *L*, whose period is $T_{f}=2W_{fg}$, corresponding to the period of the electrode arrangement for the fine measurement.

For a more general form, Equation (2) can be further formulated as
(5)$$C_{A}\left(x\right)=C_{f}\left(x+\phi_{f}\right)+a$$
where $\phi_{f}$ is the initial phase of displacement, which means that the displacement starts at a certain location.

The function $C_{B}\left(x\right)$ can be derived in the same way. The difference in the location of grating-pattern group A and grating-pattern group B is $W_{fg}$, thus:(6)$$C_{B}\left(x\right)=C_{f}\left(x+W_{fg}+\phi_{f}\right)+b\;\;=-C_{f}\left(x+\phi_{f}\right)+b$$
where $b=A_{f}+C_{B}^{0}$. $C_{B}^{0}$ is the parasitic capacitance. Its value might be a little different from that of $C_{A}^{0}$.

Similarly, functions $C_{C}\left(x\right)$ and $C_{D}\left(x\right)$ can be derived, as follows:(7)$$\{\begin{array}{c} C_{C}\left(x\right)=+C_{f}\left(x-T_{f}/4+\phi_{f}\right)+c\\ C_{D}\left(x\right)=-C_{f}\left(x-T_{f}/4+\phi_{f}\right)+d \end{array}$$
where $c=A_{f}+C_{C}^{0}$, $d=A_{f}+C_{D}^{0}$

The structure’s design makes sure that the functions $C_{E}$(*x*)~$C_{H}\left(x\right)$ for the coarse measurement have no more than a single cycle, which ensures the uniqueness of measurements within the displacement measurement range *L*. When $C_{E}$(*x*)~$C_{H}\left(x\right)$ have just one cycle, whose period is $T_{c}=L$, they can be expressed as:(8)$$\{\begin{array}{l} C_{E}\left(x\right)=+C_{c}\left(x+\phi_{c}\right)+e\\ C_{F}\left(x\right)=-C_{c}\left(x+\phi_{c}\right)+f\\ C_{G}\left(x\right)=+C_{c}\left(x-T_{c}/4+\phi_{c}\right)+g\\ C_{H}\left(x\right)=-C_{c}\left(x-T_{c}/4+\phi_{c}\right)+h \end{array}$$
where $e=A_{c}+C_{E}^{0}$, $f=A_{c}+C_{F}^{0}$, $g=A_{c}+C_{G}^{0}$, $h=A_{c}+C_{H}^{0}$, and
(9)$$C_{c}\left(x\right)=\{\begin{array}{l} -\frac{2A_{c}}{W_{c}}x+A_{c}\\ -A_{c}\\ \frac{2A_{c}}{W_{c}}x-A_{C}\left(3+\frac{2I_{c}}{W_{c}}\right)\\ A_{c} \end{array}.$$

In Equation (9), $A_{c}=\frac{\varepsilon W_{c}L_{c}}{2d}$ is the amplitude of the functions for the coarse measurement and $\phi_{c}$ is the initial phase of displacement, which is related to the arrangement of the electrodes and the starting position of the displacement.

Figure 1b shows the schematic curves of functions $C_{A}$(*x*)~$C_{D}\left(x\right)$ for the fine measurement, and Figure 1d shows the schematic curves of functions $C_{E}$(*x*)~$C_{H}\left(x\right)$ for the coarse measurement. It can be seen that the two curves in each pair $C_{A}\left(x\right)$ and $C_{B}\left(x\right)$, $C_{C}\left(x\right)$ and $C_{D}\left(x\right)$, $C_{E}\left(x\right)$ and $C_{F}\left(x\right)$, and $C_{G}\left(x\right)$ and $C_{H}\left(x\right)$ have the opposite phase. This demonstrates that $C_{f}\left(x+\phi_{f}\right)$ and $C_{f}\left(x-T_{f}/4+\phi_{f}\right)$ are orthogonal to each other and $C_{c}\left(x+\phi_{c}\right)$ and $C_{c}\left(x-T_{c}/4+\phi_{c}\right)$ are orthogonal to each other.

## 3. Modification of the Measurement Signal Model

In practical applications, the capacitance signal values that are generated by the measuring system slightly deviate from the theoretical values due to such factors as fabrication errors, installation errors, and environmental disturbances. In Figure 2, we illustrate some different types of fabrication errors (Figure 2a–c) and installation errors (Figure 2d,e). Figure 2a shows errors in the dimensions and position of electrodes. These fabrication errors may be caused by the limitations to the processing methods and the equipment’s accuracy. Position error $e_{1}$ will change the signal orthogonality. Position error $e_{2}$ expresses the non-uniformity in the spacing between two adjacent electrodes; this error influences the periodic consistency of signals. An error in the size of electrodes is denoted $e_{3}$; this error will lead to variation in the amplitude with a change in the displacement of a periodic capacitance signal. Although the dielectric film surfaces of the MOVER and the STATOR will always keep in contact when they slide, gap ***d*** will slightly vary because of such factors, including non-uniformity in the dielectric film’s thickness and deformation of the substrate. The thickness of the dielectric film that is coated on an electrode might not be uniform due to limitations to the processing methods, and Figure 2b shows such a case. Figure 2c shows the deformation of the substrate that might be caused by internal stress or an external mechanical force. Variation in gap ***d*** will result in not only a change in amplitude in the same periodic signal but also differences in the amplitude among different signals, which will also make the signals nonlinear. Figure 2d shows that there is a rotation angle between the MOVER and the STATOR, and an installation error can result in a difference in amplitude and a phase error in different measurement signals. Figure 2e illustrates the misalignment error between the MOVER and the STATOR. This installation error can be avoided by making the length $L_{fg}$ of the electrodes on the MOVER larger than the length $L_{f}$ of those on the STATOR. Besides fabrication errors and installation errors, environmental disturbances, such as mechanical vibrations and electromagnetic interference, can also have an impact on measurement signals.

The above analyses show that fabrication errors, installation errors, and environmental disturbances bring about the problems of changes in amplitude, non-orthogonality, nonlinearity, and noise in the measurement signals. Thus, the model of measurement signals can be modified, as follows:(10)$$C_{p}^{\prime }=C_{p}^{\alpha }f_{p}+N\;,\;p=A,B,\ldots ,H.\;$$

In Equation (10), $C_{p}^{\alpha }$ (p=A,B,…H) are the measurement signal functions that were modified with non-orthogonality. $f_{A}$~$f_{H}$, which are called scaling coefficient functions, were introduced to modify the signal model due to changes in the amplitude and signal nonlinearity. *N* is the noise that was added to the measuring signal model and it represents $N_{f}^{m}+N_{f}$ (*p* = A, B, C, D) or $N_{c}^{m}+N_{c}$ (*p* = E, F, G, H). $N_{f}^{m}$ and $N_{c}^{m}$ are high frequency noises that were introduced into the signals for the fine measurements and the coarse measurements due to mechanical vibrations and interference from electromagnetic signals, respectively. $N_{f}$ and $N_{c}$ are the white noises that were also added to the signals for the fine measurements and the coarse measurements, respectively [51]. Figure 3 illustrates the simulated curves of the signal functions before and after modification (further information regarding the modified signal model can be found in the Appendix A).

## 4. Signal Processing Method

In general, displacement resolution can be expressed as:(11)$$\begin{array}{c} \delta =\frac{\Delta x}{\Delta y/\sigma_{N}}\\ \;\;\;=\Delta x/\eta \end{array}$$
where $\eta =\Delta y/\sigma_{N}$ is defined as the signal resolution, Δy is defined as the maximum variable signal, Δx is the change in displacement corresponding to Δy, and $\;\sigma_{N}$ is the standard deviation of noise *N*. For the measuring signal model $C_{A}^{\prime }$~$C_{D}^{\prime }$, we have $\Delta x=\Delta x_{f}=W_{f}$ and $\Delta y=\Delta C_{f}=2A_{f}$. For the measuring signal model $C_{E}^{\prime }$~$C_{H}^{\prime }$, we have $\Delta x=\Delta x_{c}=W_{c}$ and $\Delta y=\Delta C_{c}=2A_{c}$.

Signal processing methods were investigated to reduce noise ($\sigma_{N}$) and increase sensitivity (Δy/Δx) and resolution (δ or η). Sensitivity (Δy/Δx) can also be expressed as the maximum variable signal Δy, because Δx is a constant. Three signal processing methods, namely the differential method, the ratio method, and the differential–ratio method, were analyzed, based on the modified measurement signal model established above (Equation (10)). Table 1 presents the definitions for the signal processing methods, and Figure 4 shows the simulated curves of fine signals and coarse signals. For each method, we discuss the maximum variable signal, noise, and displacement resolution in detail (more information can be found in the Appendix A).

Table 2 compares the simulation results from the three signal processing methods. Signal processing significantly improved the signal resolutions of both fine signals and coarse signals. The signal resolutions from the differential method are much higher than those from the other two methods. The signal resolutions of fine signals that were processed with the ratio method being six to seven times worse than those that were processed with the differential method. The signal resolutions of coarse signals that were processed with the ratio method are three to four times worse than those that were processed with the differential method. The signal resolutions from the ratio method and the differential–ratio method are in the same order of magnitude. As demonstrated in Figure 4b, the simulated curves that are based on the ratio method are obviously distorted and they have lost their original characteristics. The signal resolutions were recalculated with $N_{f}^{m}=0$ and $N_{c}^{m}=0$, and those from the differential and differential–ratio methods were almost the same (Table 3).

## 5. Displacement Calculation Model

In Figure 4, there are regions where the signals are insensitive to changes in displacement at the peaks and troughs of both fine signals and coarse signals, and these regions, which are called dead zones, cannot be used to measure the displacement. The two fine signal curves are orthogonal to each other and the two coarse signal curves are also orthogonal. By selectively and alternately using the two orthogonal signal curves, the dead-zone regions can be avoided when measuring displacement. Our displacement calculation model includes a coarse calculation model and a fine calculation model. Displacement, as calculated by the coarse calculation model, provides absolute position information. With the assistance of the absolute position information, the fine calculation model can be used to calculate high-precision displacement through two fine signals. The displacement calculation model can be given as:(12)$$x=f\left(y_{c1},y_{c2},y_{f1},y_{f2}\right).$$

We take the simulated curves of the coarse signals from the differential method as an example to aid the analysis. As illustrated in Figure 5, parameters $y_{0}$, $y_{H}$, and $y_{L}$ were determined to help segment the coarse signal curves and build a coarse signal calculation model [16]. Point A and point B are the intersection points of the two coarse signal curves, as shown in Figure 5a. The $y=y_{0}$ reference line was selected under the condition of $y_{c2}\left(x=0\right)\leq y_{0}\;\leq y_{c2}\left(x=L\right)$. The $y=y_{L}$ reference line was selected in the case of $max\left\{min\left(y_{c1}\right),min\left(y_{c2}\right)\right\}<y_{L}<y_{c2}\left(x=x_{A}\right)$. The $y=y_{H}$ reference line was chosen according to $y_{c2}\left(x=x_{B}\right)<y_{H}<min\left\{max\left(y_{c1}\right),max\left(y_{c2}\right)\right\}$. The coarse signal curves were divided into five linear parts and then used to calculate the coarse signal displacement: DD’, EH, JK, NO, and RQ. We adopted an interval of overlap between the two adjacent linear parts to ensure that the calculation was reliable; one of the overlaps is marked in Figure 5a. As shown in Figure 5b, with $y_{c1}\geq y_{H}$ and $y_{c2}\leq y_{0}$, the linear part DD’ of $y_{c2}$ was fitted into function $f_{1}^{c}\left(y_{c1},y_{c2}\right)$ to calculate the coarse displacement ($x_{c}$). Under the condition of
$y_{L}\leq y_{c1}<y_{H}$ and $y_{c2}<0$, the linear part EH of $y_{c1}$ was fitted into function $f_{2}^{c}\left(y_{c1},y_{c2}\right)$ to compute the coarse displacement, as shown in Figure 5c. As demonstrated in Figure 5d, with $y_{c1}<y_{L}$, the linear part JK of $y_{c2}$ was fitted into function $f_{3}^{c}\left(y_{c1},y_{c2}\right)$ to determine the coarse displacement. As illustrated in Figure 5e, under the condition of $y_{L}\leq y_{c1}<y_{H}$ and $y_{c2}>0$, the linear part NO of $y_{c1}$ was fitted into function $f_{4}^{c}\left(y_{c1},y_{c2}\right)$ to identify the coarse displacement. As demonstrated in Figure 5f, when $y_{c1}\geq y_{H}$ and $y_{c2}\geq y_{0}$, the linear part RQ of $y_{c2}$ was fitted into function $f_{5}^{c}\left(y_{c1},y_{c2}\right)$ to resolve the coarse displacement.

According to the above analyses, the coarse calculation model can be described. as follows:(13)$$x_{c}=\{\begin{array}{ll} f_{1}^{c}\left(y_{c1},y_{c2}\right) & y_{c1}\geq y_{H}\;\mathrm{and}\;y_{c2}\leq y_{0}\\ f_{2}^{c}\left(y_{c1},y_{c2}\right) & y_{L}\leq y_{c1}<y_{H}\;\mathrm{and}\;y_{c2}<0\\ f_{3}^{c}\left(y_{c1},y_{c2}\right) & y_{c1}<y_{L}\\ f_{4}^{c}\left(y_{c1},y_{c2}\right) & y_{L}\leq y_{c1}<y_{H}\;\mathrm{and}\;y_{c2}>0\\ f_{5}^{c}\left(y_{c1},y_{c2}\right) & y_{c1}\geq y_{H}\;\mathrm{and}\;y_{c2}\geq 0 \end{array}.$$

Although the five intervals DD’, EH, JK, NO, and RQ of the coarse signal curves are linear in theory, there remains slight nonlinearity due to many factors. Thus, in Equation (13), $f_{i}^{c}\left(y_{c1},y_{c2}\right)$(*i* = 1, 2, 3, 4, 5) was fitted to the polynomial function and the degrees of the polynomial were adjusted according to the fitting residuals. The total number of functions in the coarse calculation model was five in this case and this number might be higher or lower if the phase of the coarse signals is different or there is less than a single cycle in the coarse signals.

As shown in Figure 6, the fine signal curves of 1.5***T_f_*** in length were selected to help establish the fine calculation model. Each cycle of the fine signal curves was evenly divided into four parts, including two linear parts for measurement and two nonlinear parts with dead-zone regions to be avoided [16]. The linear part of signal curve $y_{f1}$ in interval AB is expressed by polynomial function labeled $f_{1}^{f}\left(y_{f1},y_{f2}\right)$. The linear part of signal curve $y_{f2}$ in interval BC is expressed by a polynomial function, labeled $f_{2}^{f}\left(y_{f1},y_{f2}\right)$, and so on. Except for the first linear part and the last linear part, the interval length of linear part is $T_{f}/4$. The fine calculation model is described as:(14)$$x_{f}=f_{n}^{f}\left(y_{f1},y_{f2}\right),\;n=\;1,\;2,\;\ldots$$
where *n* is the identification number, which is obtained by:(15)$$n=⎡\frac{x_{c}+T_{f}/4-x_{0}}{T_{f}/4}⎤.$$

In Equation (15), symbol ⎡ ⎤ is an up-rounding operator and $x_{0}$ is a constant that represents the initial phase of displacement of the fine signal curves. According to Equations (12)–(15), the displacement calculation model is expressed, as follows:(16)$$x=f\left(y_{c1},y_{c2},y_{f1},y_{f2}\right)=f_{⎡\frac{f_{i}^{c}\left(y_{c1},y_{c2}\right)+T_{f}/4-x_{0}}{T_{f}/4}⎤}^{f}\left(y_{f1},y_{f2}\right).$$

In Equation (16), when $y_{c1}\geq y_{H}$ and $y_{c2}\leq y_{0}$, *i* = 1; when $y_{L}\leq y_{c1}<y_{H}$ and $y_{c2}<0$, *i* = 2; when $y_{c1}<y_{L}$, *i* = 3; when $y_{L}\leq y_{c1}<y_{H}$ and $y_{c2}>0$, *i* = 4; and, when $y_{c1}\geq y_{H}$ and $y_{c2}\geq y_{0}$, *i* = 5.

## 6. Analysis of Error in the Displacement Calculation Model

In this section, we analyze the error in the displacement calculation model. Here, ‘error’ refers to the difference between the displacement calculated while using the displacement calculation model and the displacement from a reference calibration system. Such a kind of errors is also called mapping error [5]. These errors are systematic errors of the capacitive displacement measuring system itself, and some of these errors can be compensated for by means of corrections. Within the displacement range, for any displacement position $x^{0}$, we have the corresponding signals $y_{f1}^{0}$, $y_{f2}^{0}$, $y_{c1}^{0}$, and $y_{c2}^{0}$, and the identification number $n^{0}$. According to Equations (13)–(15), the coarse displacement, identification number, and fine displacement can be calculated, respectively, as follows:(17)$$x_{c_{cal}}^{0}=f_{i}^{c}\left(y_{c1}^{0},y_{c2}^{0}\right)$$
(18)$$n_{cal}^{0}=⎡\frac{x_{c_{cal}}^{0}+T_{f}/4-x_{0}}{T_{f}/4}⎤$$
(19)$$x_{f_{cal}}^{0}=f_{n_{cal}^{0}}^{f}\left(y_{f1}^{0},y_{f2}^{0}\right).$$

The final error in the displacement calculation model is
(20)$$\Delta =x^{0}-x_{f_{cal}}^{0}.$$

According to Equation (18), the possible values of $\left(n^{0}-n_{cal}^{0}\right)$ are –1, 0, or 1 because the coarse displacement error is $\Delta x_{c}^{0}=x^{0}-x_{c_{cal}}^{0}$. When the value of ($n^{0}-n_{cal}^{0})\;\mathrm{equals}\;0$, we have $\Delta =x^{0}-f_{n^{0}}^{f}\left(y_{f1}^{0},y_{f2}^{0}\right)$. In this case, signal $y_{f1}^{0}$ is only in the linear part of the curve $y_{f1}^{d}$ and the fine displacement is calculated by the fine calculation function $f_{n^{0}}^{f}\left(y_{f1},y_{f2}\right)$ (Figure 7b). Thus, the final error Δ is determined by the function $f_{n^{0}}^{f}\left(y_{f1},y_{f2}\right)$ and the error component that is caused by nonlinearity can be compensated for by properly adjusting the degrees of the fine calculation functions. The result of the error analysis is the same when the value of ($n^{0}-n_{cal}^{0})$
equals 1 or –1. Figure 7c shows the case of $n_{cal}^{0}=n^{0}-1$. In this case, the fine displacement should have been calculated by the fine calculation function $f_{n^{0}}^{f}\left(y_{f1},y_{f2}\right)$ while using signal $y_{f1}^{0}$, but it was actually calculated by the fine calculation function $f_{n^{0}-1}^{f}\left(y_{f1}^{0},y_{f2}^{0}\right)$ using signal $y_{f2}^{0}$ because of the coarse displacement error $\Delta x_{c}^{0}$. However, the final error Δ$=x^{0}-f_{n^{0}-1}^{f}\left(y_{f1}^{0},y_{f2}^{0}\right)$ will usually be large, because signal $y_{f2}^{0}$ is beyond the interval range of the function $f_{n^{0}-1}^{f}\left(y_{f1},y_{f2}\right)$. At the boundaries of the two adjacent fine calculation functions, the bigger the coarse error $\Delta x_{c}^{0}$ is, the bigger the final error Δ might be. To reduce the error Δ, the interval length can be increased to make sure that signal $y_{f2}^{0}$ is in the “internal” interval of the fine calculation function $f_{n^{0}-1}^{f}\left(y_{f1}^{0},y_{f2}^{0}\right)$. After expanding the interval length from $T_{f}/4$ to $T_{f}\left(1+2\varepsilon \right)/4$, the coarse error $\Delta x_{c}^{0}$ should satisfy $\left|\Delta x_{c}^{0}\right|\ll \varepsilon T_{f}/4$. Figure 6 shows the expanded interval length. However, it is important to note that ε should be as small as possible to avoid as much nonlinearity as possible in the fitting section of the fine calculation functions.

The error in the displacement calculation model was determined by the error in the fine calculation functions based on the above analyses. It was found that the model might be locally affected by the error in the coarse calculation functions. The overall uncertainty in the displacement calculation model was determined by the maximum error in the fine calculation functions. The coarse displacement model and the fine displacement model are both composed of several functions, and these functions are independent of each other. Any two adjacent coarse calculation functions or fine calculation functions will have overlapping intervals. The characteristics of independence and overlapping intervals of these functions ensure that the error in the model is not affected by differences in the signal amplitude, non-orthogonality, and signal offset. Adjusting the degrees of the coarse calculation functions and the fine calculation functions can compensate for the nonlinearity in signals.

## 7. Experiments and Discussions

A prototype was fabricated to verify the sensing principle of the proposed displacement measuring system. The MOVER and the STATOR of the measuring system were fabricated by a micromachining method. A kind of wafer glass, called BF33, was selected to be the substrate, and gold was chosen to be the electrode material that was deposited on the glass substrate. The periodical width of the fine grating-pattern groups that were located in the center was 400 um with a ***W_f_*** of 160 um and an ***I_f_*** of 40 um. That of the coarse grating-pattern groups on the sides was 9.9 mm with a ***W_c_*** of 3.96 mm and an ***I_c_*** of 0.99 mm. A layer of Si_3_N_4_ with a thickness of ∼500 nm was sputtered onto the surfaces of the electrodes as the dielectric film, due to its excellent properties, such as thermal stability, corrosion resistance, low density, high hardness, and a low friction coefficient [52,53,54,55]. The MOVER was able to move relative to the STATOR in the contact mode with the dielectric film.

Figure 8 shows the results of an adhesion strength test between the Si_3_N_4_ and base material that was performed while using a Nano-Scratch tester. Adhesion strength was considered to be the normal critical load when the Si_3_N_4_ film started to exfoliate and break away from the base material. It can be seen from Figure 8a and Table 4 that the adhesion strength between the Si_3_N_4_ and the BF33 substrate is ∼51.48 mN. The value is ∼14.66 mN between the Si_3_N_4_ and the gold material, which is shown in Figure 8b and Table 4. The adhesion strengths meet the use requirements.

As shown in Figure 9, the planarity of the MOVER and the STATOR was obtained while using a zygo Nexview™ white-light interferometer. The three-dimensional surface morphology of the STATOR and the MOVER can be seen in the white-light interference map (Figure 9a,c). As illustrated in Figure 9b, the height difference in the STATOR substrate is only 0.533 um over the span of 17 mm. In Figure 9d, the height difference in the MOVER substrate is 0.629 um over the span of 13 mm. The deformation of the substrate might have been caused by the high temperature and internal stress that occur during the manufacturing process. A very small change in the gap was caused by substrate deformation when the MOVER moved relative to the STATOR in the contact mode. As mentioned above, the scaling coefficient functions were introduced to modify gap changes in the measuring signal model.

Experiments were carried out to calibrate the displacement measuring system, which helped to establish the displacement calculation model and test the measuring system’s performance. Figure 10a illustrates the overall experimental setup; more detail is shown in Figure 10b. The entire experimental setup was placed on an active vibration isolation platform in a clean room and the room temperature was maintained at ∼20 °C. The STATOR and the MOVER were mounted on mechanical parts to guide the MOVER to move relative to the STATOR in the contact mode. The pushrod, which was mounted on a high-accuracy (less than ±100 nm) motorized positioning system, drove the MOVER to move relative to the STATOR. A HEIDENHAIN-CERTO length gauge with high accuracy (less than ±0.03 μm) was used to calibrate the displacement measuring system. The signal processing unit was used to measure signals from the displacement measuring system at a maximum sampling rate of ∼3600 Hz. A kind of Capacitance-to-Digital Converter (CDC) in signal processing unit was used to converts capacitive information into digital signal by counting number of discharging-time increment of a resistance-capacitance (RC) circuit. The relationship between the measured digital signal value and displacement was established through calibration parameters. To keep simplicity, some analyses of experimental results were in the level of measured digital signal values before being converted so that some physical quantities are dimensionless. Nevertheless, the analyses were capable to provide sensitive information of displacement measuring system by considering the linear components of the calibration parameters.

Figure 11a shows the experimental signal-displacement curves of the displacement measuring system with a range of 5 mm. The parasitic capacitance of the four signals for the fine measurement was ∼43,000 units and the amplitude $A_{f}$ was ∼21,000 units. The four signals for the coarse measurement had a parasitic capacitance of ∼30,000 units and an amplitude $A_{c}$ of ∼21,500 units. The curves reflect differences in the signal amplitude among the fine signals or the coarse signals and variations in the signal amplitude with changes in displacement. Figure 11a also present the non-orthogonality of signals.

The same conclusion as in the theoretical analyses can be drawn that only the curves that were obtained from the ratio method show obvious distortion and lose their original characteristics, according to the experimental graphs of the fine signals and coarse signals from the three different signal processing methods (Figure 11b–d). Table 5 lists the maximum variable signal (Δy), signal noise ($\sigma_{N}$), and signal resolution (η), which show the differences in the three signal processing methods. The resolutions of the fine signals are approximately 2–3 times higher than those of the original signals; however, the resolutions of the coarse signals do not show much improvement. The ratio method still cannot be applied due to the distortion and bending although the resolutions of the two fine signals from the ratio method are improved. There is not a large difference between the resolutions of the fine signals from the differential method and those of the fine signals from the differential–ratio method, which might account for the low-intensity, high-frequency noise. According to the signal resolutions in Table 5 and Equation (11), the displacement resolutions from the differential and differential–ratio methods are approximately 4.8 nm and 5.9 nm, respectively.

As seen in Figure 11b or Figure 11d, the amplitudes of the two fine signals or the two coarse signals slightly change with the displacement, and there is also a small difference in the amplitude between the two fine signals or the two coarse signals. The two fine signals or two coarse signals are not orthogonal to each other due to small phase errors, and there also exist offset errors in the fine or coarse signals. The error in the model is not affected by differences in signal amplitude, non-orthogonality, and signal offset, as shown in Section 6. Thus, the overall uncertainty in the displacement calculation model error is less than ±40 nm (Figure 11e).

As illustrated in Figure 12a, the overall uncertainty in the coarse calculation model is ±13.5 um when the fitting degree is 3 and the overlapping interval ε is 0.1. The overall uncertainty in the fine calculation model with a degree of 7 and an overlapping interval ε of 0.1 is ±31 nm, which can be seen in Figure 12b. Figure 12c shows the calculated displacement through the established displacement calculation model with a coarse degree of 3, a coarse overlapping interval of 0.1, a fine degree of 7, and a fine overlapping interval of 0.1. The calculated displacement is unique over the whole displacement range of 5 mm. The overall uncertainty in the displacement calculation model is ±34 nm, which is slightly larger than that in the fine displacement model (±31 nm), as demonstrated in Figure 12d.

Another displacement calculation model that differed only in fine degree was established using the same experimental data in order to verify the influence of fine degree on the error in the model. The uncertainty in the model with a fine degree of 5 is ±60 nm (Figure 12e), nearly twice as much as that shown in Figure 12d. A displacement calculation model that only differed in the overlapping interval was also established to determine the effect of overlapping interval on the error in the model. The overall uncertainty of ±36 nm is slightly larger than that of ±34 nm; however, there are several local points with very large error (Figure 12f).

Even when the local maximum error in the coarse calculation model reached ±17 um (Figure 12a), it did not affect the uncertainty in the displacement calculation model for tens of nanometers (Figure 12d). The error in the displacement calculation model mainly depends on the error in the fine model and the error in the coarse calculation model locally affects it. The overall uncertainty in the displacement calculation model can be improved by properly adjusting the fine degree and the fine overlapping interval. The fact that the error in the displacement calculation model was a little larger than that in the fine calculation model might be due to the fact that it is also affected by the error in the coarse calculation model.

## 8. Conclusions

This work describes a novel displacement measuring system that is based on capacitive grating and is capable of obtaining absolute position information with both high accuracy and a long range. The contact working mode between the MOVER and the STATOR minimizes the gap changes. The dead-zone problem was solved through the configuration of two orthogonal signals. The two orthogonal periodic coarse signals are used to provide absolute displacement information while using a simple method, and the two orthogonal periodic fine signals are further used to obtain a highly accurate displacement measurement with the help of absolute position information.

Fabrication errors, installation errors, and environment disturbances were analyzed, and the measurement signal model was modified according to these analyses. Three signal processing methods were also analyzed, and the results indicate that signal sensitivity and signal resolution can be effectively improved. An analysis of the error in the displacement calculation model showed that the maximum error in the fine calculation functions determined the overall displacement uncertainty and it might be locally affected by the error in the coarse calculation functions. The error in the displacement calculation model was found to not be affected by the differences in signal amplitude, non-orthogonality, and signal offset. Adjusting the degrees of the coarse functions and the fine functions can compensate for signal nonlinearity.

The experimental results are consistent with the results of the abovementioned theoretical analyses. The experimental results show that the adhesion strengths between Si_3_N_4_ and the base material meet the use requirements. The three-dimensional morphology of the STATOR and the MOVER showed that the substrate suffered very small deformation. The experiments confirmed the conclusions that were drawn from the results of the signal processing method and error analyses. A measuring system with a range of 5 mm was used to show that the displacement resolution and the error in the displacement calculation model could reach ±4.8 nm and ±34 nm, respectively. The experiments and the theoretical analyses both indicate that the proposed measuring system has great potential for achieving an accuracy of tens of nanometers and a range of hundreds of millimeters.

## Figures and Tables

**Figure 1 sensors-19-05339-f001:**
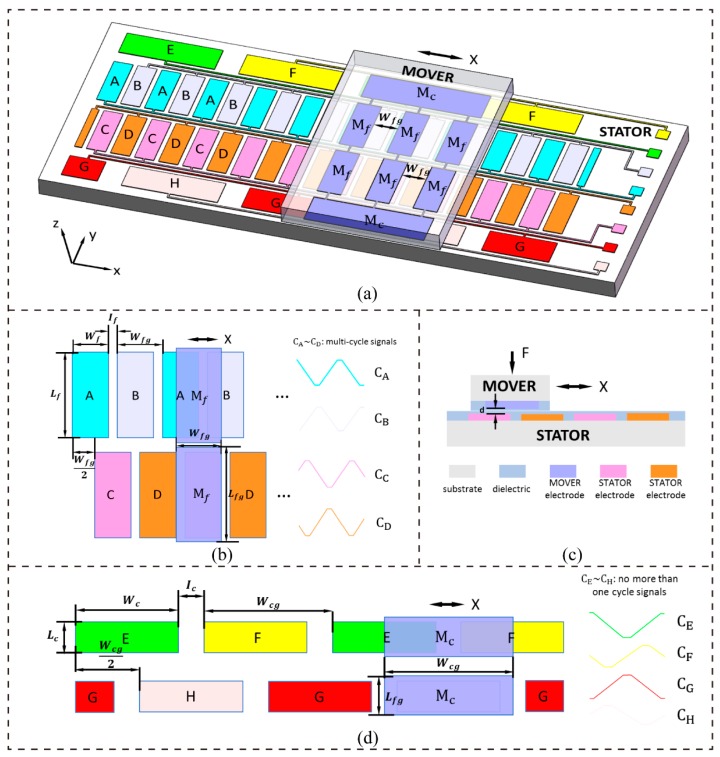
Schematic of the displacement measuring system. (**a**) Overall structure of the displacement measuring system; (**b**) top view of fine displacement measurement; (**c**) section view of the displacement measuring system; and, (**d**) top view of coarse displacement measurement.

**Figure 2 sensors-19-05339-f002:**
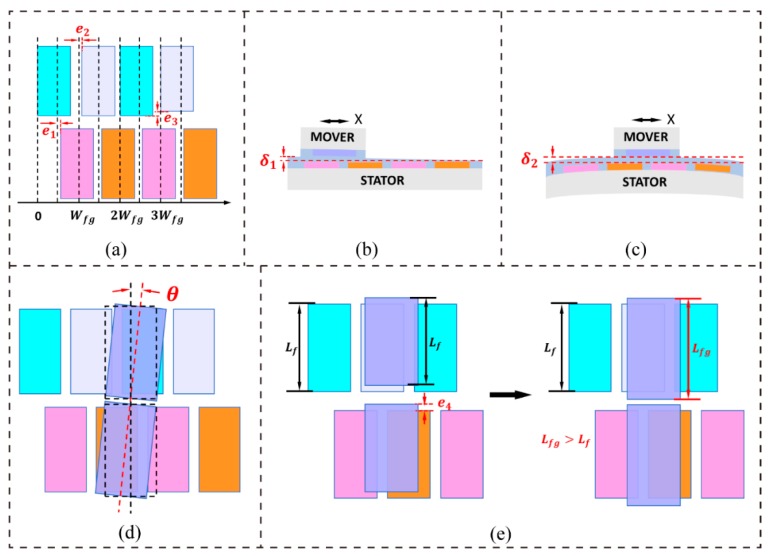
Illustrations of different types of fabrication and installation errors. (**a**) Errors in the dimensions and position of electrodes; (**b**) non-uniformity in the dielectric film’s thickness; (**c**) substrate deformation; (**d**) rotation error between the MOVER and the STATOR; and, (**e**) a misalignment error between the MOVER and the STATOR.

**Figure 3 sensors-19-05339-f003:**
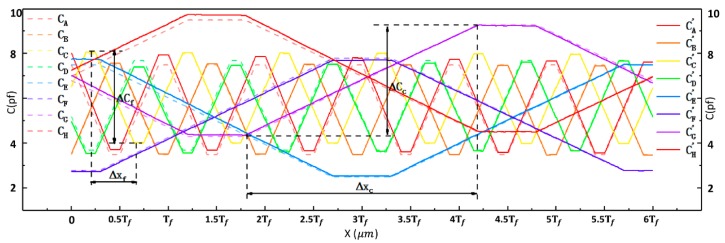
Simulated curves of the signal functions before and after modification.

**Figure 4 sensors-19-05339-f004:**
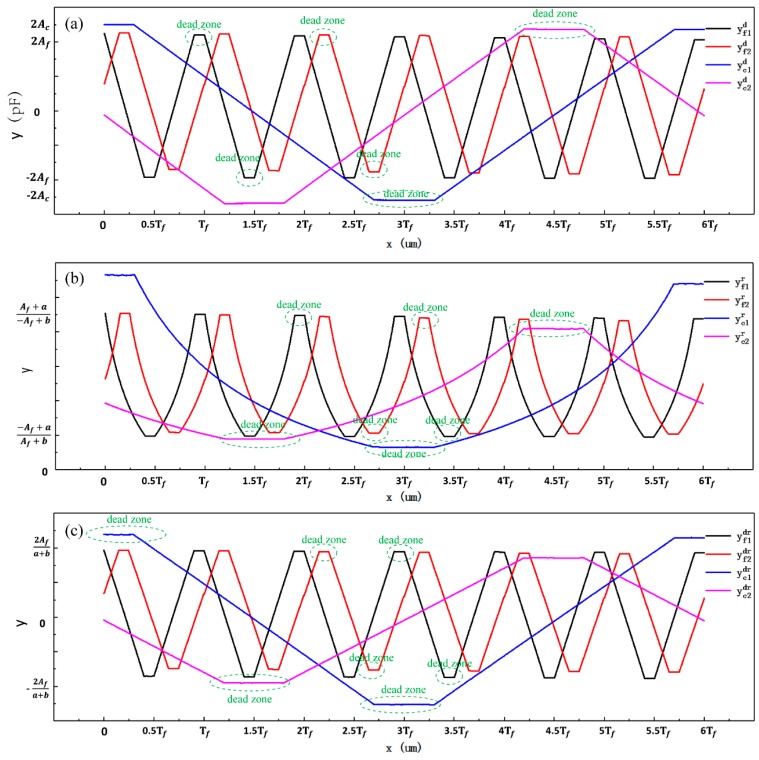
Simulated curves of fine signals and coarse signals obtained by three different processing methods. (**a**) curves obtained by the differential method; (**b**) curves obtained by the ratio method; and, (**c**) curves obtained by the differential–ratio method.

**Figure 5 sensors-19-05339-f005:**
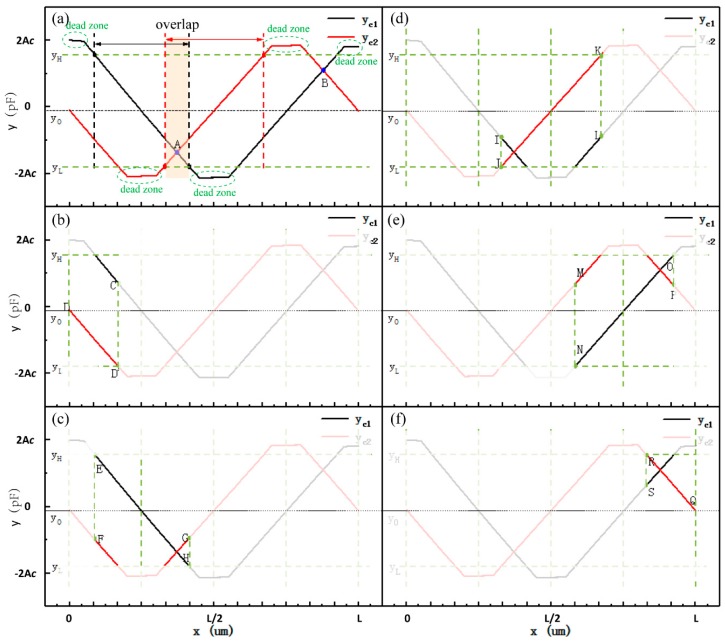
Schematic diagrams of the coarse calculation model. (**a**) the selection of $y_{0}$, $y_{H}$, and $y_{L}$; (**b**) the schematic diagram of the signal fitting interval DD’; (**c**) the schematic diagram of the signal fitting interval EH; (**d**) the schematic diagram of the signal fitting interval JK; (**e**) the schematic diagram of the signal fitting interval NO; and, (**f**) the schematic diagram of the signal fitting interval RQ.

**Figure 6 sensors-19-05339-f006:**
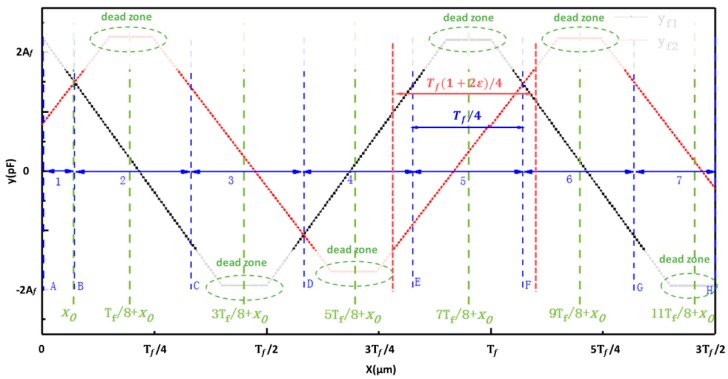
A schematic diagram of the identification number for the fine calculation model.

**Figure 7 sensors-19-05339-f007:**
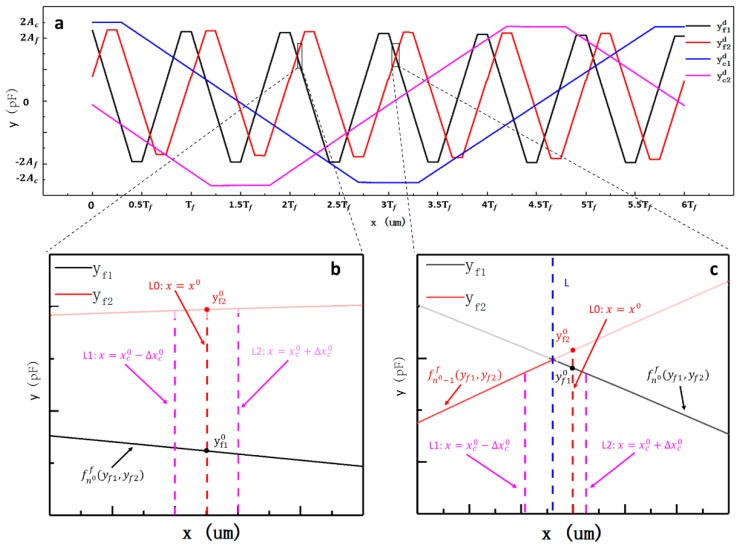
Schematic diagrams of the analysis of error in the displacement calculation model. (**a**) The schematic diagram of the signal processed by the differential method; (**b**) the schematic diagram of the error analysis under the condition of $n^{0}-n_{\mathrm{cal}}^{0}=0$; and, (**c**) the schematic diagram of the error analysis under the condition of $n^{0}-n_{\mathrm{cal}}^{0}=1$.

**Figure 8 sensors-19-05339-f008:**
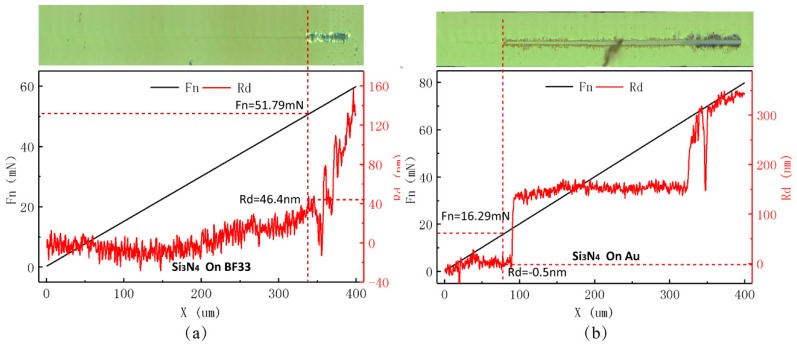
Adhesion strength test between Si_3_N_4_ and base material performed using a Nano-Scratch tester. (**a**) Adhesion strength between Si_3_N_4_ and the BF33 substrate; and, (**b**) Adhesion strength between Si_3_N_4_ and the gold material.

**Figure 9 sensors-19-05339-f009:**
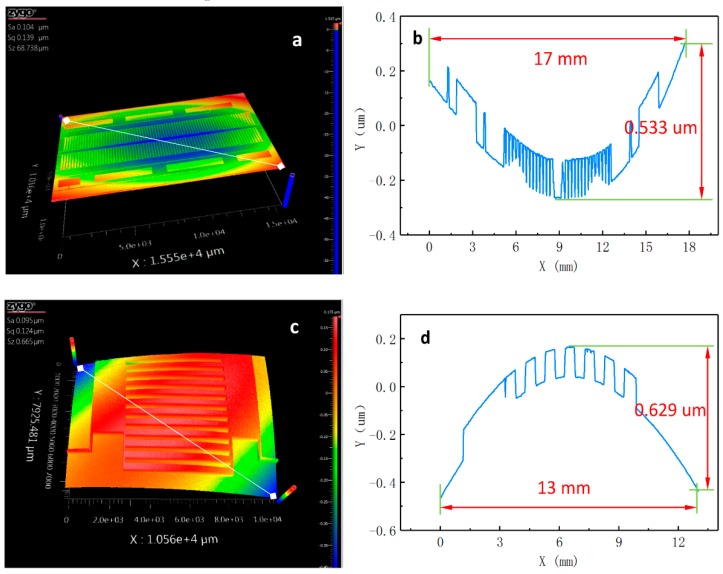
The planarity of the MOVER and the STATOR was obtained by using a zygo Nexview™ white-light interferometer. (**a**) Three-dimensional morphology of the STATOR; (**b**) height difference in the STATOR substrate; (**c**) three-dimensional morphology of the MOVER; and, (**d**) height difference in the MOVER substrate.

**Figure 10 sensors-19-05339-f010:**
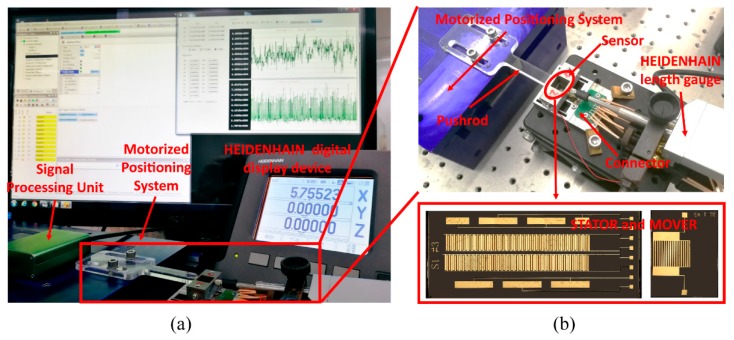
(**a**) A photograph of the experimental setup; and, (**b**) the displacement measuring system in detail.

**Figure 11 sensors-19-05339-f011:**
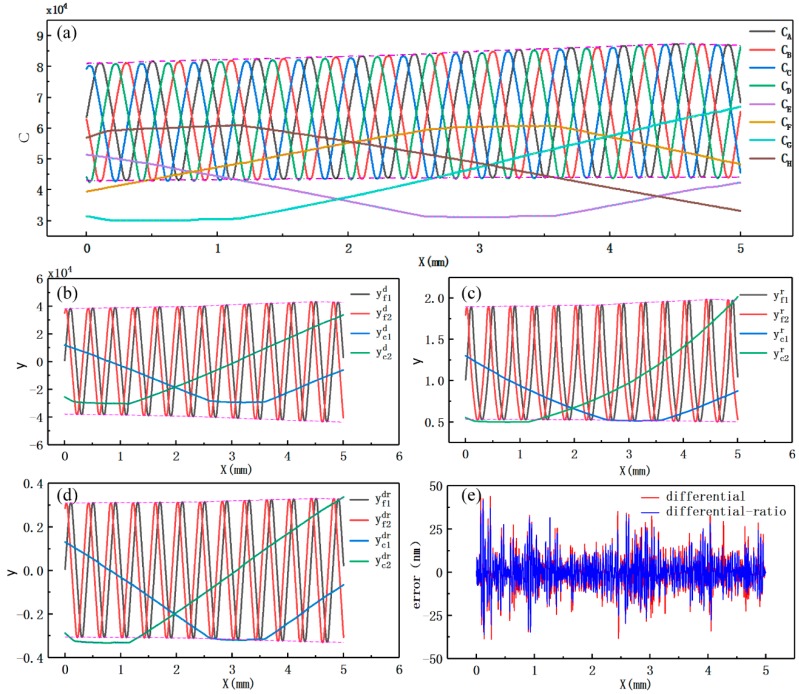
Experimental signal-displacement curves. (**a**) Original signal-displacement curves; (**b**) signal curves obtained from the differential method; (**c**) signal curves obtained from the ratio method; (**d**) signal curves obtained from the differential–ratio method; and, (**e**) the error in the displacement calculation model when using the differential method and the differential–ratio method.

**Figure 12 sensors-19-05339-f012:**
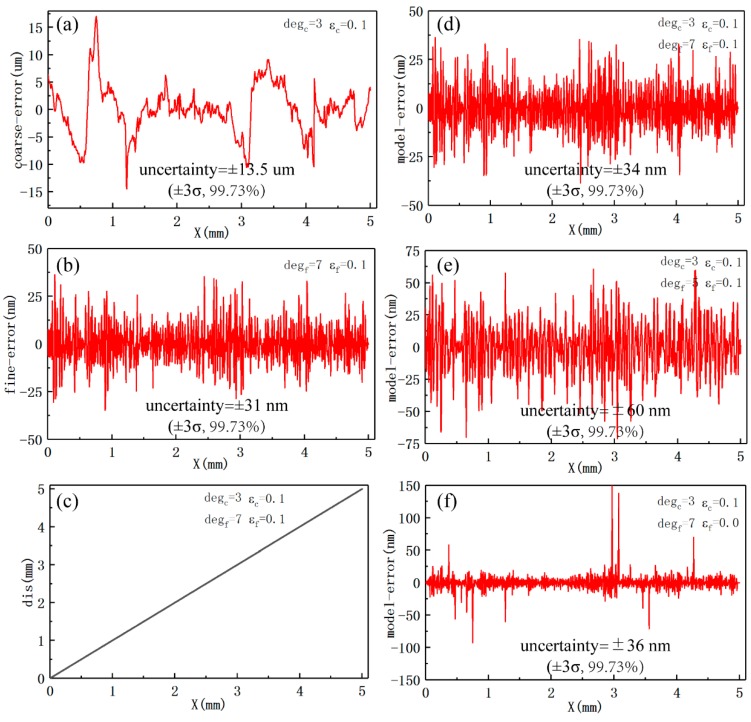
Results of experiments with the measuring system. (**a**) The error in the coarse calculation model with a degree of 3 and an overlapping interval of 0.1; (**b**) the error in the fine calculation model with a degree of 7 and an overlapping interval of 0.1; (**c**) the calculated displacement over the whole displacement range; (**d**) the error in the displacement calculation model with a coarse degree of 3, a coarse overlapping interval of 0.1, a fine degree of 7, and a fine overlapping interval of 0.1; (**e**) the error in the displacement calculation model with a coarse degree of 3, a coarse overlapping interval of 0.1, a fine degree of 5, and a fine overlapping interval of 0.1; and, (**f**) the error in the displacement calculation model with a coarse degree of 3, a coarse overlapping interval of 0.1, a fine degree of 7, and a fine overlapping interval of 0.

**Table 1 sensors-19-05339-t001:** Definitions for signal processing methods.

	Differential Method	Ratio Method	Differential–Ratio Method
Fine Signals	$\{\begin{array}{c} y_{f1}^{d}=C_{A}^{\prime }-C_{B}^{\prime }\\ y_{f2}^{d}=C_{C}^{\prime }-C_{D}^{\prime } \end{array}$	$\{\begin{array}{c} y_{f1}^{r}=C_{A}^{\prime }/C_{B}^{\prime }\\ y_{f2}^{r}=C_{C}^{\prime }/C_{D}^{\prime } \end{array}$	$\{\begin{array}{c} y_{f1}^{dr}=\frac{C_{A}^{\prime }-C_{B}^{\prime }}{C_{A}^{\prime }+C_{B}^{\prime }}\\ y_{f2}^{dr}=\frac{C_{C}^{\prime }-C_{D}^{\prime }\;}{C_{C}^{\prime }+C_{D}^{\prime }} \end{array}$
Coarse Signals	$\{\begin{array}{c} y_{c1}^{d}=C_{E}^{\prime }-C_{F}^{\prime }\\ y_{c2}^{d}=C_{G}^{\prime }-C_{H}^{\prime } \end{array}$	$\{\begin{array}{c} y_{c1}^{r}=C_{E}^{\prime }/C_{F}^{\prime }\\ y_{c2}^{r}=C_{G}^{\prime }/C_{H}^{\prime } \end{array}$	$\{\begin{array}{c} y_{c1}^{dr}=\frac{C_{E}^{\prime }-C_{F}^{\prime }}{C_{E}^{\prime }+C_{F}^{\prime }}\\ y_{c2}^{dr}=\frac{C_{G}^{\prime }-C_{H}^{\prime }}{C_{G}^{\prime }+C_{H\;}^{\prime }} \end{array}$

**Table 2 sensors-19-05339-t002:** The maximum variable signal (Δy), signal noise ($\sigma_{N}$), and signal resolution (η) from different signal processing methods by numerical simulation ^1^.

		Δy	$\sigma_{N}$	η			Δy	$\sigma_{N}$	η
Fine	$C_{A}∼C_{D}$	4.0000	4.1 × 10^−3^	976	Coarse	$C_{E}∼C_{H}$	5.0000	6.5 × 10^−3^	770
	$y_{f1}^{d}$	8.0000	4.7 × 10^−4^	17,112		$y_{c1}^{d}$	10.0000	1.8 × 10^−3^	5650
	$y_{f2}^{d}$	8.0000	4.8 × 10^−4^	16,754		$y_{c2}^{d}$	10.0000	1.7 × 10^−3^	5739
	$y_{f1}^{r}$	1.6762	6.4 × 10^−4^	2609		$y_{c1}^{r}$	2.3586	1.8 × 10^−3^	1292
	$y_{f2}^{r}$	1.6427	5.8 × 10^−4^	2820		$y_{c2}^{r}$	2.2294	7.4 × 10^−4^	3013
	$y_{f1}^{dr}$	0.7273	2.0 × 10^−4^	3579		$y_{c1}^{dr}$	0.9709	4.7 × 10^−4^	2066
	$y_{f2}^{dr}$	0.6838	1.8 × 10^−4^	3912		$y_{c2}^{dr}$	0.7246	2.7 × 10^−4^	2734

^1^ The parameters Δy, σN, and η are dimensionless.

**Table 3 sensors-19-05339-t003:** Signal resolutions from the differential and differential–ratio methods when $N_{f}^{m}=0$ and $N_{c}^{m}=0$.

Method	$\eta_{y_{f1}}$	$\eta_{y_{f2}}$	$\eta_{y_{c1}}$	$\eta_{y_{c2}}$
Differential	17,116	16,896	5612	5644
Differential–Ratio	16,180	16,244	5308	5324

**Table 4 sensors-19-05339-t004:** Adhesion strength between Si_3_N_4_ and base material (unit: mN).

Base Material	1	2	3	Average Value
BF33	51.79	50.58	52.06	51.48
Au	16.29	15.01	12.68	14.66

**Table 5 sensors-19-05339-t005:** The maximum variable signal (Δy), signal noise ($\sigma_{N}$), and signal resolution (η) with different signal processing methods. ^1^

Fine		Δy	$\sigma_{N}$	η	Coarse		Δy	$\sigma_{N}$	η
	$C_{A}\&C_{B}$	44,378	4.3	10,441		$C_{E}\&C_{F}$	20,741	3.9	5318
	$C_{C}\&C_{D}$	44,454	3.3	13,309		$C_{G}\&C_{H}$	32,342	2.6	12,391
	$y_{f1}^{d}$	86,487	2.1	41,381		$y_{c1}^{d}$	41,447	3.9	10,654
	$y_{f2}^{d}$	86,578	1.9	45,567		$y_{c2}^{d}$	64,159	2.5	25,561
	$y_{f1}^{r}$	1.4782	4.1 × 10^−5^	36,498		$y_{c1}^{r}$	0.7882	1.2 × 10^−4^	6853
	$y_{f2}^{r}$	1.4815	2.7 × 10^−5^	55,905		$y_{c2}^{r}$	1.5179	4.2 × 10^−5^	36,487
	$y_{f1}^{dr}$	0.6582	1.8 × 10^−5^	36,770		$y_{c1}^{dr}$	0.4521	5.3 × 10^−5^	8498
	$y_{f2}^{dr}$	0.6604	1.9 × 10^−5^	34,041		$y_{c2}^{dr}$	0.6707	3.0 × 10^−5^	22,062

^1^ The parameters Δy, σN, and η are dimensionless.

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
