# Peer review of "Error Analysis and Modeling for an Absolute Capacitive Displacement Measuring System with High Accuracy and Long Range"

_sensors, 2019, doi:10.3390/s19245339_

Round 1

Reviewer 1 Report

Please find a native English colleague who will be able to understand the technical context and help you express it effectively in English. YOU must edit this work.

technical comment: sections 6. Relate the measurement uncertainty numerical analysis using modeling of and actual measurements. Define the types of measurement uncertainty and address them.

Reviewer 2 Report

The article focuses on a novel absolute capacitive displacement system and in this regard some details are proposed about its modeling and error analysis.

The topic is very interesting and the manuscript is quite well written, nevertheless some issues suggest that your paper should be improved:

Page 1 line 37 (pdf file, Introduction). I suggest changing “challenge” to “challenging” Page 1 line 39. The term “major types” is not clear. Try “most common transducers” instead. Page 2 line 63. I suggest changing “efficient” to “efficiently”. Page 2 line 82. Check the text “model is modified based on basic measuring…”: it is not clear. Page 2 line 85. I suggest changing “. Displacement calculation” to “. The displacement calculation”. Other “the” have been missed in the text: please, check it. Page 2 line 85. I suggest changing “to get high accuracy absolute displacement…” to “to get high accuracy in absolute displacement”. Please, follow the International Vocabulary of Metrology (VIM) for a better (not) use of the term “accuracy”. Although in your title the first words are “Error Analysis”, the term “uncertainty” is never used in your paper: it is a lack that should be filled. In this regard I suggest improving the terminology (i.e. use the VIM) as well as the experimental approach in your error analysis. Page 3 line 98 (pdf, Basic Measuring Principle). I suggest changing “…are ???, ??? , and…” to “…are ??? and ??? respectively, and…” Page 3 line 99. I suggest changing “…and ??? .” to “…and ??? …” Page 6 line 182 (pdf, Modification of Measurement Signal Model). In the text “?? and ?? are white noises also added…” you talk about “white noise”: please, justify it and/or refer to some literature. Page 6 line 195 (pdf, Signal Processing Method). In the text “To reduce noise and increase sensitivity and SNR, signal processing methods are investigated.” the three quantities “noise“, “sensitivity” and “SNR” deserve more details and should be clearly defined as the quality of your experimental results depend on their evaluation indeed. Moreover results about the above quantities should be clearly reported (together with the corresponding measurement units). Page 7 line 207. The quantity “signal resolution“ deserve more details and should be clearly defined. Page 8 line 215. The measurement units should be reported in the table. Please, check missing measurement units in the other tables, too. Page 12 line 337-338 (pdf, Experiments and Discussions). You should provide/evaluate the metrological characteristics (e.g. in terms of expanded uncertainty) of the WHOLE experimental setup. Moreover a table about the characteristics of the HEIDENHAIN-CERTO length gauge may be of help. Page 14 line 362-363. “Displacement resolutions from differential and differential-ratio methods are about 4.8 nm and 5.9 nm, respectively.” It is not clear how you get this result, please give some more details. Page 14 line 367-368. “However, the error of measuring system is still less than ±40 nm (Figure 11e), and system accuracy is not affected by the amplitude error, phase error and DC error.” More details should be provided about the error and its evaluation: you should evaluate the measurement uncertainty at a specific confidence level (referring to diagrams is not enough). Moreover your “±40 nm” error seems to me too close to the uncertainty of your reference gauge (i.e. the HEIDENHAIN-CERTO length gauge) but, on the other hand, I haven’t found details about the environmental conditions of your setup (i.e. temperature, humidity) that can have a significant impact on your measurement results (at a nm scale even a breath may provide significant dilatations). Another critical issue is about the connections among your sensor, the motorized positioning system and the H. length gauge: what about them and their influence on your results?! Have you considered them in your error analysis? Please, refer to the GUM (https://www.bipm.org/en/publications/guides/gum.html ). On the basis of the above considerations, I suggest you to check your results and conclusions.

Reviewer 3 Report

The content and research seems to be generally ok, but the text must be thoroughly reviewed for the English language and many of the figures are not or hardly readable. Further detailed comments are given in the annotated text.  

Round 2

Reviewer 2 Report

Although I appreciate the efforts of the authors to improve their manuscript, some concerns still remain.

In particular the uncertainty evaluation is not clear as well as the outcomes of your study and your contribution respect to other devices in literature, e.g. what is the relationship between the accuracy claimed for other devices in literature and your error analysis for the displacement calculation model of your system? Is your system more accurate or suitable than the measurement systems you referenced in your introduction?

Moreover it is not clear how is it possible that some physical quantities you used are dimensionless, i.e. variable signal and standard deviation of the noise.

Reviewer 3 Report

The paper is still more complicated than maybe necessary, mainly because of the rather crippled English at many places. I recommend that a professional editing service is asked to improve the manuscript. Technically all looks rather sound.

Round 3

Reviewer 2 Report

Please,

with the aim to make your results clearer, I suggest adding some text about the CDC use, as mentioned in your last reply:

"In our researches, a kind of Capacitance-to-Digital Converter (CDC) is used to measure the relative capacitance. The CDC is used to converts capacitive information into digital signal by counting number of discharging-time increment of a resistance-capacitance (RC) circuit. The output of CDC is integers within 24 bits, and has no physical meaning. Thus the variable signal
, signal noise and signal resolution are dimensionless. The relationship between the measured signal value and displacement is established through calibration parameters. To keep simplicity, analysis is in the level of measured signal values before being converted so that some physical quantities are dimensionless."
